# Interventions to reduce stigma related to contraception and abortion: a scoping review

Annik Sorhaindo, Ulrika Rehnstrom Loi 

UNDP-UNFPA-UNICEF-WHO-World Bank Special Programme of Research, Development and Research Training in Human Reproduction (HRP), Department of Sexual and Reproductive Health and Research, World Health Organization, 20 Avenue Appia, 1211 Geneve, Switzerland

**Correspondence to**
Dr Ulrika Rehnstrom Loi; rehnstromu@who.int

## ABSTRACT

**Objectives** We conducted a scoping review to identify the types, volume and characteristics of available evidence and analyse the gaps in the knowledge base for evaluated interventions to reduce contraception and abortion stigma.

**Design** We conducted a search of five electronic databases to identify articles published between January 2000 and January 2022, and explored the websites of relevant organisations and grey literature databases for unpublished and non-commercial reports. Articles were assessed for eligibility, and data were extracted.

**Data sources** We searched MEDLINE, PubMed, Embase, Web of Science and PsycINFO.

**Eligibility criteria** Articles included were: (1) published between January 2000 and January 2022, (2) written in English, (3) reports of the evaluation of an intervention designed to reduce contraceptive and/or abortion stigma, (4) used any type of study design and (5) conducted in any country context.

**Data extraction and synthesis** Included studies were charted according to study location, study aim, study design, type of contraceptive method(s), study population, type of stigma, and intervention approach.

**Results** Some 18 articles were included in the final analysis (11 quantitative, 6 qualitative and 1 mixed methods). Fourteen of the studies focused exclusively on abortion stigma, and two studies focused on contraception stigma only; while two studies considered both. A majority of the studies aimed to address intrapersonal stigma. We found no interventions designed to address stigma at the structural level. In terms of intervention approaches, seven were categorised as education/training/skills building, five as counselling/peer support, three as contact and three as media.

**Conclusion** There is a dearth of evaluations of interventions to reduce contraception and abortion stigma. Investment in implementation science is necessary to develop the evidence base and inform the development of effective interventions, and use existing stigma scales to evaluate effectiveness. This scoping review can serve as a precursor to systematic reviews assessing the effectiveness of approaches.

## INTRODUCTION

Health-related stigma has important implications for medical, social and psychological well-being of those affected.[1–3] In reproductive health, stigmatising attitudes towards

## STRENGTHS AND LIMITATIONS OF THIS STUDY

⇒ The systematic inclusion of contraceptive stigma in the search exposed a gap in the evidence of interventions designed to address this reproduction health stigma.

⇒ The search strategy developed was designed to capture a broad swath of the literature for review.

⇒ The interventions were organised according to an existing framework for abortion stigma interventions

⇒ We limited our search to studies that were published in English which may have excluded key articles written in other languages.

⇒ We focused on studies that had an element of evaluation, thereby excluding descriptions of activities taking place to reduce contraception and abortion stigma without efforts to measure effectiveness, acceptability or implementation.

people receiving or providing care is mediated by perceived alignment of practices surrounding human reproduction with societal morals and expectations.[4 5] Contraception and abortion stigma have a range of health and non-health related consequences, such as unintended pregnancy, perceptions of promiscuity, criminalisation, secrecy, unsafe abortion and associated complications and poor quality of abortion care.[6 7] Contraceptive use stigma emerges where there is censure of the control of fertility, and where contraceptive use conflicts with social norms regarding the life course timing and role of sexual activity, and desired family size.[7–9] Although rates vary by country,[10] abortion is a routine component of sexual and reproductive health. Abortion stigma creates challenges and barriers to access to safe services that can result in deleterious physical, economic and social consequences for women.[11 12] The two phenomena are rooted in the negation of the right of inidviudals to decide whether and when to have children, make decisions about and have control of their body and sexuality and manifest similarly in social norms and

practices that hinder access to family planning and lead to unintended pregnancy.[7]

Abortion stigma has been defined as, '…a negative attribute ascribed to women who seek to terminate a pregnancy that marks them, internally or externally, as inferior to the ideals of womanhood'.[13] Abortion stigma is located and manifests in various aspects of society and affects a wide range of groups.[14] Kumar and colleagues[13] adapted a framework[15] to illustrate the levels of society at which abortion stigma operates and where interventions are needed. At the governmental/structural level abortion stigma is thought to operate in educational, economic, legal and health policies that reflects social norms discrediting abortion. Internal procedures and guidelines that perpetuate abortion stigma characterise the institutional level. At the community level, abortion stigma influences social factors, attitudes and actions that cause individuals involved with abortion to lose status and social networks. Individual-level stigma, or the internalisation of negative feelings, such as shame and guilt associated with the experience of abortion, manifests as intrapersonal and interpersonal stigma.[13] Investigation of contraceptive stigma is limited; however, one analysis with adolescents in Kenya determined that negative stereotypes and discrimination results in delays in access to healthcare and missed opportunities for contraceptive counselling.[7 16]

Eliminating stigma in sexual and reproductive healthcare is viewed as critical to avoid negative outcomes and for overall public health.[17] A recent scoping review explored the ways in which abortion stigma influences quality in abortion care[18] and determined that stigma reducing interventions are important for improving quality and decreasing unsafe abortion. A systematic review of multilevel health stigma reduction interventions concluded that research should expand the groups targeted.[19] Although, interventions to reduce contraception and abortion stigma exist,[20] the volume, approaches employed, levels targeted and effectiveness of these remain largely unknown.

We undertook a scoping review to identify the key characteristics of this evidence base and analyse gaps. A scoping review approach is appropriate for such indications[21 22] and is a first step to be followed by assessments of effectiveness in systematic review. The aim of this scoping review was to identify the types, volume and characteristics of available evidence and analyse gaps in the knowledge base for interventions to reduce contraception and abortion stigma.

## METHODS

This review was conducted in accordance with Arksey and O'Malley's scoping review methodology and Levac and colleagues' methodological enhancement.[23 24] We report the methodology and findings according to PRISMA (Preferred Reporting Items for Systematic Reviews and Meta-Analyses) guidelines[25] (online supplemental file 1).

### Search terms

To locate literature for this scoping review, we developed a search strategy with support from experienced librarians in the WHO library (online supplemental file 2). We conducted this search in five electronic databases: MEDLINE, PubMed, Embase, Web of Science and PsycINFO to identify articles published between January 2000 and January 2022. We used the Covidence tool (Covidence Systematic Review Software, Veritas Health Innovation, Melbourne, Australia, available at www.covidence.org) to screen, review, organise and extract information from the articles.

### Inclusion criteria

We used the following eligibility criteria for inclusion in the scoping review: (1) published between January 2000 and January 2022, (2) written in English, (3) reports on

| **Table 1** | Categories of analysis |
| --- | --- |
| **Category** | **Definition** |
| Type of stigma[13 15] | |
| Intrapersonal | Internalisation of negative feelings, such as shame and guilt, associated with experience with abortion. |
| Interpersonal | Direct or enacted forms of stigma, such as verbal harassment and/or physical assault based on association with abortion. |
| Community | Social factors and attitudes and actions that cause individuals involved with abortion to lose status in their communities and social networks. |
| Organisational/institutional | Procedures and guidelines internal to organisations that perpetuate abortion stigma. |
| Governmental/structural | Educational, economic, legal, health policies and other regulation that reflects social norms that discredit abortion. |
| Intervention approach[20] | |
| Counselling/peer support | Provide healthcare information, emotional support and empowerment to people who experience stigma. |
| Education/training/skills building | Provision of information about the stigmatised group and their concerns with the goal of reducing stigma. |
| Contact | In-person interactions that can lead to reductions in prejudice between groups. |
| Protest | An action that publicly calls attention to stigmatising attitudes and/or behaviours that promote these attitudes. |
| Social marketing | A strategy that uses marketing techniques to achieve behavioural and/or health-related goals. |
| Media | Extended forms of contact that may involve fictional and non-fictional media or visualisations. |

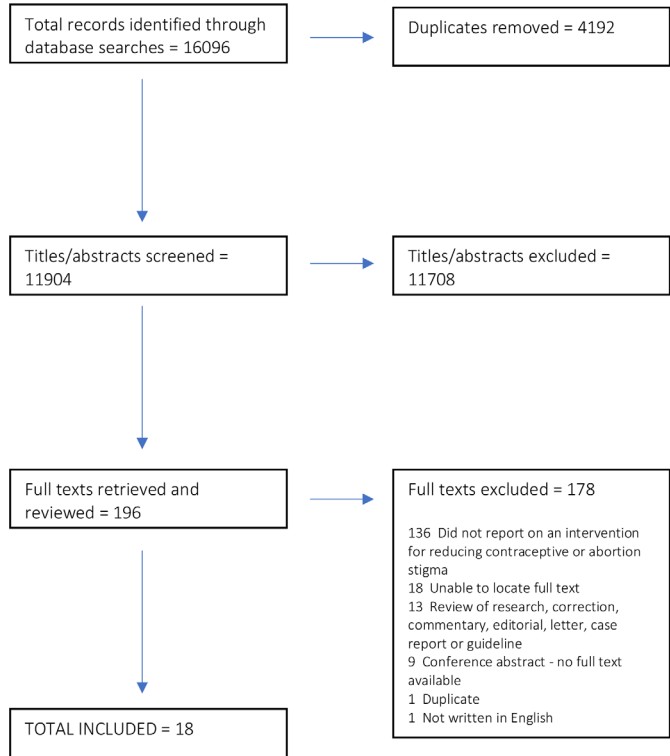

**Figure 1** Scoping review PRISMA flowchart. PRISMA, Preferred Reporting Items for Systematic Review and Meta-Analysis.

the evaluation of an intervention specifically designed to reduce contraceptive and/or abortion stigma in any relevant population as part of the primary aim of the study, (4) uses any type of study design and (5) was conducted in any country context.

### Exclusion criteria
We excluded titles and abstracts if they were a review of other research,[22] correction, commentary, editorial, letter, case report or guideline. Furthermore, we excluded articles that were unavailable through institutional holdings.

### Data extraction
Two researchers independently screened the titles and abstracts against the inclusion and exclusion criteria. The full texts deemed relevant by both reviewers were retrieved and reviewed. Two researchers independently assessed the full-text articles to confirm eligibility and reconcile any discrepancies. We additionally conducted a search for grey literature of evaluations of interventions to reduce contraceptive and abortion stigma by exploring the websites of key organisations working in sexual and reproductive health and databases designed to extract unpublished literature, but located no relevant texts during this process.

### Data charting process and analysis
We developed a data-charting form to extract relevant information to answer our scoping review question. One researcher classified and charted all studies according to

study location, study aim, study design, type of contraceptive method(s), study population, type of stigma and intervention approach . The second researcher reviewed 20% of the extractions of included articles to identify any discrepancies. Both researchers met to discuss, re-review articles and resolve any conflicts.

For the analysis of the included literature, we drew an existing stigma intervention framework outlined in a 2013 White Paper[20 26] including the following categories: counselling approaches, education and training (information and skills-building), social marketing, protest and contact with affected groups. Although designed to address abortion stigma, the framework drew evidence from a range of health stigmas and thus may also apply to contraceptive stigma. The interventions included in the four categories aim to address intrapersonal and interpersonal stigma (table 1). The White Paper included an additional three categories for interventions designed to address stigma at the structural level: institutions, laws and policies and the media.[20] We used this overall framework to organise the literature included in this scoping review. Finally, study quality was not assessed, as this is not a requirement for scoping reviews.[22 23]

### Patient and public involvement
We did not engage with patients or the public during the development of this review.

### Findings
Our initial search resulted in 16 096 records. After removing 4192 duplicate articles, two researchers independently screened 11 904 titles and abstracts against the inclusion and exclusion criteria (figure 1). The full texts of the 196 titles and abstracts were retrieved and reviewed. Of these, 178 were excluded (figure 1). The remaining 18 texts were included in this review (figure 1).

### Characteristics of included articles
The 18 included studies were conducted in 10 countries: Cambodia (1), Colombia (1), India (1), Indonesia (1), Kenya (1), Mexico (1 study was conducted in Mexico City) (3), South Africa (1), Spain (1) and the USA(8). One study included unspecified countries in the Latin America and Caribbean region and sub-Saharan Africa.[27] Of the 18 included studies, 11 reported on a quantitative evaluation, 6 reported on a qualitative evaluation and 1 used mixed methods approaches. Fourteen of the studies focused exclusively on abortion stigma, and two studies focused on contraception stigma. The remaining two studies considered both. A majority of the studies aimed to address intrapersonal stigma. We found no interventions designed to address stigma at the structural level. Online supplemental file 3 presents the full list of included studies.

### Description of stigma reduction interventions
Of the 18 included papers, 7 were categorised as education/training/skills building, 5 were categorised as counselling/peer support, 3 as contact and 3 as media (online

supplemental file 3). Each intervention is described by the category below.

### Education/training/skills building

Values clarification is a training intervention for addressing thoughts, feelings and attitudes towards issues that are emotive, such as contraception and abortion. The first article, published in 2015, was an evaluation of the effectiveness of the *Difficult Dialogues* strategies in reducing negative attitudes among the midwives in South Africa.[28] The qualitative study suggested that the intervention had some influence on increasing participants' empathy for their patients.[28] The second article evaluated the TEACH (Teaching Everything About Contraceptive Health) programme which aimed to dispel myths about contraception and abortion among obstetrician–gynaecology residents in a faith-based hospital in the USA. The results of the evaluation suggested that the intervention had limited impact on reducing discriminatory attitudes towards patients seeking abortion services.[29] Finally, a recent evaluation of another values clarification workshop conducted with residents in Catholic hospitals in the USA found that, when presented with abortion-related scenarios, residents held less stigmatising attitudes.[30]

Researchers in Spain implemented an intervention for nursing students including a simulation of a family planning counselling session. Using a quasi-experimental design, the evaluation determined that 2 weeks after the intervention, students' attitudes about contraceptive methods improved as measured by relevant validated scales.[31]

An evaluation of an intervention conducted in India aimed to empower women to seek services and to normalise abortion and the provision of support by the community and family members of women who seek services. Via a quasi-experimental approach, a scale was used to assess attitudes towards abortion following the intervention. The results suggested that it increased perceived levels of social support for abortion within their families and self-efficacy around family planning and abortion. However, perceptions of support for abortion within communities were lower after the intervention.[32]

The Family Planning Balanced Counselling Strategy (FP-BCS) is an intervention designed to support the uptake of modern contraceptive methods. Researchers evaluated the effectiveness of the FP-BCS on attitudes, subjective norms and intentions to use modern contraception in Indonesia. An evaluation using a quasi-experimental design determined a significant increase in positive attitudes towards modern contraception among mothers as a result.[33]

An evaluation of a school-based intervention assessed whether Comprehensive Sexuality Education reduced stigma related to abortion and contraceptive use in western Kenya. Among students aged 13–21 years at follow-up, stigma scores for both contraception and abortion reduced significantly among boys and girls.[7]

Researchers developed a narrative intervention that promoted cognitive restructuring and was designed to normalise the abortion experience using an animated 4-min short film. People seeking abortion viewed the animation in the waiting room before receiving care. A randomised controlled trial assessed changes to internalised stigma as a result of the intervention. At 2-week follow-up there was no significant difference in internalised stigma between intervention and control groups.[34]

### Counselling/peer support

The Story Circles Project intervention was implemented in Mexico and designed to reduce intrapersonal stigma among women who had experienced an abortion. The Story Circles Project provided a safe space for women to share their abortion stories, connect with other women and explore their abortion decision. The intervention also included information sessions to dispel myths and misinformation. The intervention significantly reduced women's feelings of isolation helped reduce stigma.[35]

Three evaluations of the Provider Share Workshop, an intervention designed for health workers involved in abortion care, were included in this review.[36–38] The workshop sessions offer providers an opportunity to discuss and explore their experiences. The 2011 pilot study found that participants reported an 'increased interpersonal connection' with the team following the workshop.[36] In 2014, the Provider Share Workshop was conducted among providers in the USA and effectiveness was evaluated using two validated scales. Although the evaluation could not show statistically significant results, the study identified a downward trend in abortion stigma overtime.[37] Finally, the Workshop was adapted and piloted in two countries in sub-Saharan Africa and Latin America. The evaluation found decreases in overall abortion stigma, but specifically in disclosure management, internalised stigma and judgement.[38]

A qualitative assessment of the role of counselling within legal abortion services offered in non-governmental organisations in Mexico City and Colombia in addressing women's concerns and fears prior to receiving abortion care found that women experienced relief, and less guilt and internalised stigma after counselling.[38]

### Contact

In 2018, researchers assessed the effectiveness of an intervention designed to reduce interpersonal abortion stigma manifesting as secrecy among women who have experienced abortion and negative reactions from women who learn about the abortion experiences of others. The intervention used the safe spaces of existing book clubs to assess whether women would self-disclose their abortion experiences in a positive face-to-face interaction. Book clubs were recruited to read and discuss the book *Choice: True Stories of Birth, Contraception, Infertility, Adoption, Single Parenthood and Abortion*. During the book club intervention, a majority of the women with abortion histories

disclosed their experience and the book 'warmed' participants feelings about abortion over time.[39]

The online platform, Lightbulbs (*Focos*, in Spanish), was designed in Mexico to provide a safe space for women to tell their abortion stories and to increase abortion visibility. In discourse analysis of the text, researchers found that most of the stories were positive and described as a 'cathartic' experience.[40]

## Media

In 2016, researchers examined the impact of the viewing of the documentary film, *After Tiller*, in reducing third trimester abortion stigma. Viewing the film appeared to challenge myths and misperceptions about abortion. However, the authors found that the medium was limited in its ability to generate new belief systems around abortion.[41]

The results of an online study of viewers on an episode of the USA medical television programme *Grey's Anatomy* which featured an abortion storyline titled 'Papa Don't Preach'. The findings suggested that the programme did not increase viewers willingness to support a friend seeking an abortion, whether medically necessary or induced.[42]

## DISCUSSION

Despite calls for the development of interventions focused on reducing reproductive health stigmas, and specifically, contraceptive and abortion stigma, there remains significant gaps in the literature.[4 16 18 19] Most articles identified were reports of evaluations of abortion stigma interventions taking place in the USA. A handful of articles focused on Latin America, two in sub-Saharan Africa, one in Asia and one in Europe demonstrating limitations of understanding of available interventions in specific regions and contexts.

The published interventions largely focused on addressing intrapersonal and interpersonal stigma. The most common intervention approach identified was education, training and skills building. The least common identified was the contact approach. There is a significant gap in the availability of interventions to reduce contraceptive and abortion stigma at the structural level and using approaches involving institutions, and laws and policies.

There exists substantial theory about intervention approaches that should have an effect on reducing contraception and abortion stigma.[1 15 17 18] Several rigorously validated scales are available for measuring changes in contraception and abortion stigma. Programme developers and evaluators should design and test interventions based on theory and measure their effectiveness using existing validated scales. Future research may synthesise the existing evidence to understand the effectiveness of specific intervention approaches.

The methodology of this scoping review has some limitations. We limited our search to studies that were published in English; this may have inadvertently excluded key articles written in other languages. We focused on studies that had an element of evaluation. This meant that we excluded descriptions of activities taking place to reduce contraception and abortion stigma without efforts to measure effectiveness, acceptability or implementation. This methodological choice may have limited our engagement with interventions currently in use, but not evaluated. It is also possible that evaluators have not published the results of studies of ineffective or inconclusive interventions.

In the design of interventions, stigma continues to be largely conceptualised as an attribute and interventions to mitigate stigma targeted to manifestations at the individual level. Any interventions aligning with this concept may inadvertently (re)generate stigma and minimise the impact of mitigation efforts. Further attention in the development of interventions targeted to the social processes that underpin stigma may have a more profound impact on attitudes and behaviours across populations.[43] Furthermore, efforts to reduce contraceptive and abortion stigma could draw a broader stigma framework and benefit from shared concepts, tools, indicators and outcomes.[44] Scoping reviews are pragmatic precursors to systematic reviews where the scope and characteristics of an area of investigation is unclear.[22] The results of this scoping review can inform efforts to develop robust systematic reviews to assess the quality and effectiveness of approaches to reduce contraceptive and abortion stigma.

**Acknowledgements** The authors would like to acknowledge the input of Kavita Kothari, WHO Librarian, Geneva, Switzerland, for her suggestions and support in refining the search strategy.

**Contributors** AS and URL conceptualised and conducted the review, including assessing literature and data synthesis. AS prepared the manuscript with intellectual contributions from URL. Both authors approved the final manuscript. The authors alone are responsible for the views expressed in this article, and they do not necessarily represent the views, decisions, or policies of the institutions with which they are affiliated. All authors that also reviewed and edited versions of the manuscript. URL is responsible for the overall content as guarantor by accepting full responsibility for the finished work and the conduct of the study. URL had access to the data, and controlled the decision to publish. All authors approved the final manuscript for publication.

**Funding** The UNDP–UNFPA–UNICEF–WHO–World Bank Special Programme of Research, Development and Research Training in Human Reproduction, a cosponsored programme executed by the WHO, funded this work.

**Disclaimer** The views expressed in this article are those of the authors and do not necessarily represent the views of, and should not be attributed to, the WHO.

**Competing interests** None declared.

**Patient and public involvement** Patients and/or the public were not involved in the design, or conduct, or reporting, or dissemination plans of this research.

**Patient consent for publication** Not applicable.

**Ethics approval** Not applicable.

**Provenance and peer review** Not commissioned; externally peer reviewed.

**Data availability statement** No data are available. No additional data available.

**ORCID iD**
Ulrika Rehnstrom Loi http://orcid.org/0000-0002-3455-8606

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
