## [Reviewer comments · BMJ Open]

ARTICLE DETAILS

TITLE (PROVISIONAL)	Interventions to reduce stigma related to contraception and abortion: a scoping review
AUTHORS	Sorhaindo, Annik; Rehnstrom Loi, Ulrika

VERSION 1 – REVIEW

REVIEWER	Miani, Celine RAND
REVIEW RETURNED	01-Jun-2022

GENERAL COMMENTS	Thank you for the opportunity to review this article. The authors conducted a scoping review to assess the volume and scope of the evidence on interventions to reduce abortion and contraception stigma. I think that this is a valuable addition to the field. The methods are sound. I have a few comments though, which hopefully will contribute to improve the manuscript. About the general aim of the article: Up until the last paragraph of the discussion, it is not clear that the authors only wanted to include studies of interventions which reported on an evaluative component. This is a limitation, as acknowledged in the discussion, but this is also a review-defining inclusion criterion that should come up much earlier in the text, and in the abstract. The focus on evaluations, if clear from the beginning, could have more naturally led to a systematic review, or a realist synthesis than to a scoping review. I would also have been more informative, since design and outcomes of the interventions would have been thoroughly discussed. Introduction: In the introduction, the second paragraph is dedicated to abortion stigma. There is no section dedicated to contraception stigma. Why? How do the two relate? Can the existing evidence on abortion stigma be transferred to contraception stigma? Also, the White Paper framework used for the analysis has been developed for abortion stigma. How does it transfer to contraception stigma? Methods: Why start the search in 2000? Is there a justification for this choice of date? Could the authors comment on the limitation to English texts? Do they believe that it has led to the exclusion of many articles (just 1 was excluded for this reason at the last stage of screening according to the PRISMA flowchart, but maybe more before?)
---

	In the second paragraph of the “data charting and analysis process”, the authors describe a framework: this paragraph does not match the table and makes it difficult to understand the framework. Are the information-based interventions referring to Protest and Social marketing in the table? (those two are not cited in the text)? Also, in the text, Institutions, Laws and the Media are also referenced to as categories of intervention, when in the table, only media is -and institutions and laws seem to be in the type of stigma. Consistency would improve readability and add value to the analysis. “Patient and public involvement”: the justification for not involving users or members of the public is not right to me. PPI can be part of review, see for example: Pollock D, Alexander L, Munn Z, Peters MD, Khalil H, Godfrey CM, McInerney P, Synnot A, Tricco AC. Moving from consultation to co-creation with knowledge users in scoping reviews: guidance from the JBI Scoping Review Methodology Group. JBI Evidence Synthesis. 2022 Apr 1;20(4):969-79. Discussion: I understand that the review did not set out to assess the effectiveness of interventions (although it might be a missed opportunity since most of the work has been done). Still, I feel that a paragraph on the likely impact of the interventions is missing in the discussion. From a rapid assessment, I have the impression that about 2/3 of the interventions had some positive impact. What can be said about that? What types of interventions seem to be more consistently effective? Such an assessment could help identify further leads for research (e.g. a systematic review of effectiveness for a specific type of intervention, etc.). Conclusion: I believe that the conclusion is not a conclusion, but a paragraph that should go in the discussion: it adds new arguments and new references. It includes statements such as “Any interventions aligning with this concept ...efforts” which cannot directly be inferred from the review but are more an extension of the scope. Something more suited to a discussion than a conclusion. Minor comments and typos: Abstract, methods: if the reports are in database, they are still “published” and not “unpublished”, right? Introduction, second paragraph: there is a “ missing to show the end of the quote. Data charting process and analysis: “affected groups” rather than “effected groups”? Same paragraph: “three categories” instead of “three categorizes”. Characteristics of included studies: add reference after Sub-Saharan Africa, so that the reader knows which study was conducted there. Education/training...: “In 2015”, change to “in 2015” In the last paragraph, “cognitive” rather than “cognative”? Discussion, first paragraph: “most articles were reports of evaluation...”. I wouldn’t say that 8/18 is most. Rephrase? Line just above: “there remains” instead of “their remains”.
--	---

REVIEWER	Love, Gillian University of Sussex
-----------------	---------------------------------------

REVIEW RETURNED	14-Jun-2022
-------------

GENERAL COMMENTS	This is an interesting and well-written scoping review. Its aims and objectives are clearly presented, the methodology is sound, and the results are also clearly presented. The fact that no eligible studies were found that addressed stigma at a structural level is a valuable finding, in particular. There is only one substantial recommendation I have for the authors. A question that came to mind while reading that was not answered is, why think of abortion and contraception stigma together? From your introduction, it is clear that abortion stigma is a relatively well established concept, but contraception stigma seems to be less so (indeed, contraception stigma isn't mentioned in the introduction after the first paragraph). I would therefore invite the authors to a) put forward a rationale for thinking these two types of stigma together, and b) make explicit the fact (if it is indeed a fact) that contraception stigma receives less attention in the literature. At the moment I am left unsure as a reader why these two phenomena are being brought together. In addition, one minor comment: Reference 17 is 'under development' – is it an appropriate reference in that case? Overall, this is an interesting and well written scoping review.
--

VERSION 1 – AUTHOR RESPONSE

Reviewer 1: Dr. Celine Miani, RAND

Reviewer 1 comment:

Thank you for the opportunity to review this article.

The authors conducted a scoping review to assess the volume and scope of the evidence on interventions to reduce abortion and contraception stigma.

I think that this is a valuable addition to the field. The methods are sound. I have a few comments though, which hopefully will contribute to improve the manuscript.

Authors' response: We are grateful to you for your positive response to our manuscript and thank you for your comments.

Reviewer 1 comment:

About the general aim of the article:

Up until the last paragraph of the discussion, it is not clear that the authors only wanted to include studies of interventions which reported on an evaluative component. This is a limitation, as acknowledged in the discussion, but this is also a review-defining inclusion criterion that should come up much earlier in the text, and in the abstract.

The focus on evaluations, if clear from the beginning, could have more naturally led to a systematic review, or a realist synthesis than to a scoping review. I would also have been more informative, since design and outcomes of the interventions would have been thoroughly discussed.

Authors' response: We agree that our focus on evaluations should be mentioned earlier in the manuscript. We have added this information to the abstract and on line 30, page 5 of the revised manuscript. We have also added comments to the end of the Discussions section suggesting further research in the form of systematic review to follow on from this work.

Reviewer 1 comment:

Introduction:

In the introduction, the second paragraph is dedicated to abortion stigma. There is no section dedicated to contraception stigma. Why? How do the two relate? Can the existing evidence on abortion stigma be transferred to contraception stigma? Also, the White Paper framework used for the analysis has been developed for abortion stigma. How does it transfer to contraception stigma?

Authors' response: Yes, there is far less research on contraceptive stigma than on abortion stigma and this is reflected in the Introduction. However, the two concepts are indeed related because at their root is the right to decide whether and when to have children, fertility regulation and have control over one's body and sexuality, which is often hampered in similar ways by social norms and practices that deny this right. Denial of the right to contraceptives and safe abortion lead to consequences in unintended pregnancy.

The White Paper referred to was designed for abortion stigma but draws upon evidence of effective approaches for reducing stigma across a range of areas of health care. Given this, we believe the approaches summarized can also apply to contraceptive stigma.

We appreciate these observations and have added language in the introduction (page 4, line 28-30) and methods (page 7, line 1-3) to clarify for readers.

Reviewers 1 comment:

Methods:

Why start the search in 2000? Is there a justification for this choice of date?

Could the authors comment on the limitation to English texts? Do they believe that it has led to the exclusion of many articles (just 1 was excluded for this reason at the last stage of screening according to the PRISMA flowchart, but maybe more before?)

Authors' response: We agreed that the bulk of articles published on interventions to reduce abortion and contraceptive stigma will have been published in the past 20 years, during which time interest in the issue grew. We also limited to English texts due to the language skills of both reviewers. We agree with the reviewer that this is a possible limitation and have added this to line 15, page 12 of the revised manuscript.

Reviewer 1 comment:

In the second paragraph of the "data charting and analysis process", the authors describe a framework: this paragraph does not match the table and makes it difficult to understand the framework. Are the information-based interventions referring to Protest and Social marketing in the table? (those two are not cited in the text)?

Also, in the text, Institutions, Laws and the Media are also referenced to as categories of intervention, when in the table, only media is -and institutions and laws seem to be in the type of stigma. Consistency would improve readability and add value to the analysis.

Authors' response: Thank you for highlighting this lack of clarity in our manuscript. We have now added the missing information from the paragraph starting on line 26 on page 6 of the revised manuscript to align the text with the information in Table 1.

Reviewer 1 comment:

“Patient and public involvement”: the justification for not involving users or members of the public is not right to me. PPI can be part of review, see for example: Pollock D, Alexander L, Munn Z, Peters MD, Khalil H, Godfrey CM, McInerney P, Synnot A, Tricco AC. Moving from consultation to co-creation with knowledge users in scoping reviews: guidance from the JBI Scoping Review Methodology Group. *JBI Evidence Synthesis*. 2022 Apr 1;20(4):969-79.

Authors’ response: We have revised the statement to the following: “We did not engage with patients or the public during the development of this review”, on line 4, page 8 of the revised manuscript.

Reviewer 1 comment:

Discussion:

I understand that the review did not set out to assess the effectiveness of interventions (although it might be a missed opportunity since most of the work has been done). Still, I feel that a paragraph on the likely impact of the interventions is missing in the discussion. From a rapid assessment, I have the impression that about 2/3 of the interventions had some positive impact. What can be said about that? What types of interventions seem to be more consistently effective? Such an assessment could help identify further leads for research (e.g. a systematic review of effectiveness for a specific type of intervention, etc.).

Authors’ response: Given that a scoping review approach does not traditionally include an assessment of intervention effectiveness, we disagree that what you suggest should be included here. Further, a systematic literature review would base any assessments of positive impact on a synthesis of the evidence. As we have not done this here, any judgements would be qualitative and potentially incorrect or misleading. We believe that it is more appropriate to keep this review within the parameters of a scoping review and leave a robust synthesis and assessment of effectiveness for a follow-up systematic literature review.

Reviewer 1 comment:

Conclusion:

I believe that the conclusion is not a conclusion, but a paragraph that should go in the discussion: it adds new arguments and new references. It includes statements such as “Any interventions aligning with this concept ...efforts” which cannot directly be inferred from the review but are more an extension of the scope. Something more suited to a discussion than a conclusion.

Authors’ response: We have added the conclusion to the discussion section.

Reviewer 1 comment:

Minor comments and typos:

Abstract, methods: if the reports are in database, they are still “published” and not “unpublished”, right?

Introduction, second paragraph: there is a “ missing to show the end of the quote.

Data charting process and analysis: “affected groups” rather than “effected groups”? Same paragraph: “three categories” instead of “three categorizes”.

Characteristics of included studies: add reference after Sub-Saharan Africa, so that the reader knows which study was conducted there.

Education/training...: “In 2015”, change to “in 2015”

In the last paragraph, “cognitive” rather than “cognative”?

Discussion, first paragraph: “most articles were reports of evaluation...”. I wouldn’t say that 8/18 is most. Rephrase?

Line just above: “there remains” instead of “their remains”.

Authors' response: Thank you for these observations. We have made all the edits as suggested.

Grey literature refers to unpublished and non-commercial reports. We have added this to the abstract.

In your comment about the use of "most" in the first line of the Discussion, we are referring to the number of articles focused on abortion, which as 16 out of the 18.

Reviewer 2: Dr. Gillian Love, University of Sussex

Comments to the Author:

This is an interesting and well-written scoping review. Its aims and objectives are clearly presented, the methodology is sound, and the results are also clearly presented.

The fact that no eligible studies were found that addressed stigma at a structural level is a valuable finding, in particular.

Authors' response: We appreciate your kind comments on our manuscripts.

Reviewer 2:

There is only one substantial recommendation I have for the authors. A question that came to mind while reading that was not answered is, why think of abortion and contraception stigma together? From your introduction, it is clear that abortion stigma is a relatively well established concept, but contraception stigma seems to be less so (indeed, contraception stigma isn't mentioned in the introduction after the first paragraph).

I would therefore invite the authors to a) put forward a rationale for thinking these two types of stigma together, and b) make explicit the fact (if it is indeed a fact) that contraception stigma receives less attention in the literature. At the moment I am left unsure as a reader why these two phenomena are being brought together.

Authors' response: Yes, that is correct. Contraceptive stigma is far less established concept than abortion stigma and this is reflected in the Introduction. We believe that the two concepts are related because at their root is the right to decide whether and when to have children, fertility regulation and make decisions about and have control over one's body and sexuality, which is often hampered in similar ways by social norms and practices that deny this right. Denial of the right to contraceptives and safe abortion lead to consequences in unintended pregnancy. We have added language in the introduction (page 4, line 28-30) and methods (page 7, line 1-3) to clarify for readers.

Reviewer 2:

In addition, one minor comment:

Reference 17 is 'under development' – is it an appropriate reference in that case?

Authors' response: This paper has now been accepted for publication and we have updated the manuscript to reflect this.

Reviewer 2:

Overall, this is an interesting and well written scoping review.

VERSION 2 – REVIEW

REVIEWER	Miani, Celine RAND
REVIEW RETURNED	21-Sep-2022
GENERAL COMMENTS	Thank you for clarifying your approach. I am looking forward to seeing the published paper.